# Investigations on the Characterization of Various Adhesive Joints by Means of Nanoindentation and Computer Tomography

**DOI:** 10.3390/ma15238604

**Published:** 2022-12-02

**Authors:** Arkadiusz Bernaczyk, André Wagenführ, Robert Zboray, Alexander Flisch, Thomas Lüthi, Birgit Vetter, Mario Rentsch, Christian Terfloth, Jörg Lincke, Tomasz Krystofiak, Peter Niemz

**Affiliations:** 1Jowat SE, 32758 Detmold, Germany; 2Institute of Natural Materials Technology, Technische Universität Dresden, 01062 Dresden, Germany; 3Center for X-ray Analytics, Empa—Swiss Federal Laboratories for Materials Science and Technology, 8600 Dübendorf, Switzerland; 4Institute of Materials Science, Technische Universität Dresden, 01062 Dresden, Germany; 5Department of Wood Science and Thermal Techniques, Poznań University of Life Sciences, 60-627 Poznan, Poland; 6Lulea University of Technology, 971 87 Lulea, Sweden

**Keywords:** wood gluing, beech, wood adhesives, adhesive joint, hardness, mechanical properties

## Abstract

The mechanical properties of cured wood adhesive films were tested in a dry state by means of nanoindentation. These studies have found that the application of adhesives have an effect on the accuracy of the hardness and elastic modulus determination. The highest values of hardness among the tested adhesives at 20 °C have condensation resins: MF (0.64 GPa) and RPF (0.52 GPa). Then the decreasing EPI (0.43 GPa), PUR (0.23 GPa) and PVAc (0.14 GPa) adhesives. The values of the elastic modulus look a little bit different. The highest values among the tested adhesives at 20 °C have EPI (11.97 GPa), followed by MF (10.54 GPa), RPF (7.98 GPa), PVAc (4.71 GPa) and PUR (3.37 GPa). X-ray micro-computed tomography was used to evaluate the adhesive joint by the determination of the voids. It has been proven that this value depends on the type of adhesive, glue quantity and reactivity. The highest values of the void ratio achieve the PUR (17.26%) adhesives, then PVAc (13.97%), RRF (6.88%), MF (1.78%) and EPI (0.03%). The ratio of the gaps increases with the higher joint thickness. A too high proportion of voids may weaken the adhesive joint.

## 1. Introduction

The use of high-performance wood adhesives has currently become essential for the production of high-demand applications, such as timber construction including GLT (Glue-laminated timber) and CLT (Cross Laminated Timber). Material properties, also on the microscale, are of great importance to improve the performance of the single constituents and the interaction in the interface regions of the composites.

Hänsel et al. (2021) summarized and discussed the current state of the art factors influencing the gluing quality of solid wood products in timber construction. The gluing of hardwood species receives special focus, as it is considered by the industry to be the most challenging process for implementation. The gluing effect is influenced by the wood, surface quality, glue, glulam structure and gluing technology. It is important in the wood selection process to consider the properties of the species, such as the grain and ring angle, density, strength, moisture behavior, chemical composition, but also wood modifications (e.g., thermal, chemical and densification). The vast majority of GLT is made of softwood, although the amount of hardwood products has been increasing recently. Hardwood, such as beech (*Fagus* sp.), ash (*Fraxinus* sp.), or birch (*Betula* sp.) are used separately or in combination with softwood species. The growing popularity of these species is hampered by the difficulty of gluing and the price of the components. Another important aspect is choosing an appropriate adhesive. The choice of adhesive type, bond line thickness, sorption and mechanical properties, adhesion to wood and aging behavior are very important. In particular, by the use of load-bearing elements, it is worth paying attention to the lamella thickness, the selection of species with the desired properties, dimensions and surface coating. Ultimately, the method of preparation, pressing conditions and the application of adhesive play an important role in the quality of adhesive bonding [1,2].

Timber construction needs to fulfill very strict requirements regarding strength, resistance to moisture and to elevated temperatures. There have been many attempts to improve these parameters with 1C-PUR (one-component polyurethane) adhesives by adding fillers (organic and inorganic) and varying the prepolymers. It has been proven that urea hard segments always have a positive effect on the thermal stability and that the adhesive indentation hardness and Young’s modulus have a direct impact on the WFP (wood failure percentage). Through the addition of filler materials, the thermal stability of the adhesives has been significantly increased. Research confirms that the mechanical properties of the 1C-PUR adhesives are significantly affected by their prepolymer composition [3].

Sebenik and Krajnc (2007) and Chattopadhyay and Webster (2009) concluded that the NCO (Isocyanate)/OH ratio and the molar mass are important factors determining the structure properties of PUR. These properties can be measured by various methods. FTIR-ATR spectroscopy and strength tests were used in the mentioned publications. It was pointed out that these parameters are very important in formulating the desired adhesive [4,5,6].

Furthermore, it is reported that a higher proportion of the still reactive, free NCO groups, a lower degree of resin polymerization and a slower reaction rate are the dominant factors for the high thermal stability (Richter et al., 2006), whereas beyond a critical free NCO content, the high stiffness of the adhesive is responsible for a decrease in the adhesion strength, These are other examples of how complicated the process is to choose the right adhesive parameters [5,6,7,8].

Clauss (2011) concluded that the polyisocyanate component is mostly responsible for the properties (elasticity, cohesion and strength) of the finished product. As a part of this research, the isocyanate (NCO) content, the crosslink density, the urethane group content and the ethylene oxide/propylene oxide ratio were varied. The author highlights the importance of the subsequent formulation (catalyst, filler, plasticizer, etc.) on the quality of the bond [9].

In formulating the composition of the additives that are often added in addition to the main ingredient additives, these can improve the properties. For example colorants (such as dyes or pigments), plasticizers (increase flexibility), and fillers. The adhesive formulation by means of additives, moreover, does not affect the mechanical properties, but is responsible for the bonding performance [10,11].

Mendoza et al. (2012) used a combined experimental–numerical modal analysis technique for determining the effective mechanical bond line properties. The numerical model analysis is relatively accurate, the difference between the respective experimental and numerical eigenmodes is less than 2.25%. This method can be used to estimate the hardness, modulus of elasticity (MOE), density of the wood and adhesive layer, and to predict the penetration of adhesives into hardwood [12]. The model was applied to beech samples joined with three different types of adhesives (PUR, UF, PVAc) under various growth ring angles, as described by Hass et al. (2010). Analyzing the results using an appropriate statistical method is crucial to interpreting the obtained results [13].

Plötze et al. (2011) determined the densities and the porosity parameters on domestic and overseas soft- and hardwoods with the application of pycnometric methods and mercury intrusion porosimetry (MIP). The hardwoods, particularly the European diffuse-porous ones, show a higher amount of micropores, which represent the microvoids or cell wall capillaries [14].

Furthermore the sorption behavior and the fluid intake was investigated as the technological characteristics in the industrial processes of impregnation and penetration of the coating materials or adhesives [15].

Adhesives with different chemical bases, absorb different amounts of water. Even low amounts of water may influence the mechanical performance of the glued wood products. Wimmer et al. (2013) performed a dynamic vapor sorption analysis to assess the sorption processes of six commercial wood adhesives. The best performing adhesive, by means of strength values, was RPF, then in decreasing order: fish glue, PUR, MUF and PVAc [16].

Furthermore, the wood strength is also moisture-dependent. Kläusler et al. (2013) noticed a decrease in strength as the moisture content increased [17]. Studies of Hass et al. (2009) showed a significant influence of growth ring angle, adhesive system and viscosity on the shear strength of the bonded elements [18].

A lot of methods for evaluating the quality of gluing and the parameters affecting the quality of gluing have been described above, which shows how important and timely this topic is.

### 1.1. Computed Tomography

The computed tomography method has been used many times to evaluate wood and the glue joints. Hass et al. (2012) investigated the penetration of adhesive and mechanical properties of adhesive bond lines by means of synchrotron radiation X-ray tomographic microscopy (SRXTM) [12].

Sanabria (2011) used limited-angle microfocus X-ray computed tomography for the glue line assessment of timber constructions. Delaminations and air gap thickness topology could be assessed for mass discontinuities larger than 150 µm between the timber lamellas. These limits did not compromise the detectability of the lamination faults for the thin glue lines of 100 to 200 µm obtained with hydraulic pressing [19].

Leggate et al. (2021) used micro X-ray computed tomography and microscopy to assess the key PUR adhesive bond criteria. There was a considerable loss in the amount of adhesive after the wet and dry test cycles for all species. There was also an extremely high frequency of voids in the glue lines for all species, which would negatively impact the bond strength and durability [20].

Bastani et al. (2016) examined the three-dimensional (3D) visualization of the penetration of PU adhesive into heat-treated Scots pine by X-ray micro-computed tomography. The 3D pattern obtained by XmCT of the PU adhesive flow in the radial direction of the heat-treated Scots pine, provided an understanding of the pathways this adhesive used (e.g., pits and lumens of the adjacent axial tracheids) to penetrate the wood [21].

Jakes et al. (2019) used X-ray computed tomography (XCT) and X-ray fluorescence microscopy (XFM) to study the adhesive flow and infiltration. The model wood–adhesive bond lines were made using loblolly pine (Pinus taeda) and BrPF adhesive [22,23]. Furthermore, synchrotron tomography was successfully used, combined with the acoustic emission to investigate the deformation of wood under tension [24].

As mentioned above, much research has been carried out with this method, but no one has yet attempted to evaluate the properties of the glue joint, based on the amount of gas void in the glued element.

### 1.2. Nanoindentation

The first determinations of the material properties of the adhesives by means of nanoindentation (NI) were recently described by Stöckel, Konnerth and Bockel [25,26,27,28,29]. Stöckel (2013) summarized and discussed the state of the art mechanical properties of pure wood adhesives. He stated that wood adhesives show a large variability of mechanical properties in the cured state. The modulus of elasticity determined by different methods covers a wide range from 0.1 GPa up to 15 GPa. The results are highly influenced by the adhesive formulation and the ambient conditions, but also by the sample preparation and by the testing method used. Modulus values of the cured wood adhesives are highest for amino resin adhesives, compared to phenolic adhesives. Adhesives based on isocyanates (polyurethanes, emulsion polyisocyanates and pMDI), epoxy resins and poly vinyl acetate adhesives represent the lower end of the range. Moisture typically causes a softening of the adhesives, whereby the phenolics and the structural amino resin show the highest susceptibility. Increased temperature typically reduces stiffness in a temperature range between 20 °C and 70 °C, in the case of polyvinyl acetate, polyurethanes, and epoxies, whereas other adhesives are less affected (e.g., condensation resins) [26].

Recently, Bockel et al. (2020) analyzed the properties of wood adhesive bonding, by means of nanoindentation, and compared it to the classical lap-shear strength. Research shows that both wood and RPF adhesive show a much lower hardness when wet. PUR adhesive does not show such a relationship [29]. The literature values for the hardness of the adhesives cover a wide range [26].

Herzele et al. (2020) investigated the impact of different chemically composed fibers on the adhesion behavior between adherend and adhesive. The less polar surface seems to be favorable for 1C-PUR, whereas surfaces of a higher polarity, e.g., pure cellulose surfaces, are less favorable for the 1C-PUR adhesives [30].

Frybort et al. (2014) investigated the polarity of wood by means of an AFM adhesion force mapping. It has shown that the polarity significantly decreases with the increasing surface age and that differences in polarity between freshly cut cell walls and native inner lumen surfaces correlate with the chemical heterogeneity, in particular the varying ratio of lignin, compared to the cell wall carbohydrates [31].

The purpose of this study was to use commercially available methods (NI and used X-ray micro computed tomography) and find a way to evaluate the adhesive bond, possibilities of characterizing and observing the relationship for various commonly used wood adhesives. It is a comparative analysis of the indicators provided by the two assessment tools and the methodologies considered.

It is expected that, especially the PUR adhesives, due to their known foaming, will have a lot of voids in the glue joint.

The advantage of these methods over those traditionally used, is their accuracy, repeatability and lack of sample destruction. The samples are measured at the nanoscale. Thanks to the fact that the samples are not destroyed during the test (as happens, for example, during strength testing) they can be reproduced. The testing of the joints is crucial, in terms of new solutions being introduced to the market. The obtained values can help adhesive manufacturers formulate new adhesive systems.

## 2. Materials and Methods

### 2.1. Materials

#### 2.1.1. Wood

The wood species used for the bonding experiments was beech (*Fagus sylvatica* L.) with a density of 750 ± 40 kg/m^3^ at a moisture content of 12.0 ± 0.5%. Beech wood was chosen due to the low content of extractives (to avoid the chemical interaction with the adhesives). The properties of beech wood, in comparison with the selected deciduous trees are summarized in Table 1.

Wood was acclimatized at 20 °C and 65% relative humidity (RH) for 30 days.

#### 2.1.2. Adhesives

Six adhesives from different manufacturers were tested.

Thin adhesive films were prepared from six different wood adhesives:

Melamine formaldehyde resin (MF), resorcinol phenol formaldehyde resin (RPF), polyvinylacetate adhesive (PVAc), emulsion-polymer-isocyanate adhesive (EPI) and one-component polyurethane adhesive (1C-PUR).

Two PUR adhesives were examined. The only difference in the recipe was the addition of the longitudinal polyamide fibers (approx. 5%) into PUR 1. The addition of the fibers was carried out by the adhesive manufacturer during the production, in a stirring process.

The parameters used during bonding are listed in Table 2. Those were selected, based on the recommendations in the technical data sheets from the manufacturers to achieve the best possible result.

##### RPF Resin

The RPF resin is based on the reaction of resorcinol with formaldehyde. In the first stage, the reaction yields linear chains. The addition of formaldehyde occurs preferably at positions 4 and 6 on the aromatic ring, while the position between the two hydroxyl groups is sterically hindered. The reaction of resorcinol with formaldehyde is influenced by the molar ratio, the concentration of the solution, the pH, the temperature and the catalyst types and alcohols used [3].

##### MF Resin

The basic reactions of the MF production consist of the methylolation and subsequent condensation. The reaction of formaldehyde with the amino groups of melamine leads to methylols, with a corresponding average degree of methylolation or distribution over the individual methylolation stages, depending on the formaldehyde excess [3].

##### PUR

PUR adhesives are based on polyadditions and polymerization reactions. Due to an excess of isocyanate, the reaction of isocyanate and polyols (polyester or polyatherpolyols) produces chains with terminal and, if necessary, lateral isocyanate groups, which can react with the moisture of the wood surfaces to be glued and thus lead to a cured system via this additional reaction. Therefore, at least one of the two surfaces must supply the amount of water required for curing, i.e., it must be porous and contain moisture.

During the curing, the reaction of the isocyanate group with the moisture produces CO_2_, which can cause the foaming of the adhesive joint [3].

##### PVAc

PVAc adhesives are physically setting adhesives. The gluing effect of PVAc is based on the removal of water by the penetration into the surface or by evaporation [3].

##### EPI

The adhesive effect is achieved by the evaporation of water from the glue line or absorption by the parts to be joined (physical setting). In addition, the chemical cross-linking takes place, which leads to a significantly higher resistance to moisture and temperature [3].

Properties of the adhesives in the Table 3 was given.

### 2.2. Methods

#### 2.2.1. Specimen Preparation

The wood samples were glued at 20 °C and 65% relative humidity.

For the nanoindentation, there were three types of samples manufactured: the adhesive film on glass, on wood and embedded in epoxy resin.

The adhesives film on the wood and glass were produced by using a film applicator with dimensions: 80 mm × 53 mm, to fit the holder in the testing machine, as shown on Figure 1a,b.

The adhesive films embedded in epoxy resin were produced using a film applicator with dimensions: 39 mm × 11 mm, to fit the holder in the testing machine, as shown on Figure 1c.

PUR adhesive films were produced using a film applicator under low humidity conditions in a dry box (<5%). The low relative humidity allows for the slow curing of the adhesive and therefore helps to reduce the formation of bubbles [7].

For the computer tomography, the samples were cut into pieces with dimensions of 6 mm × 6 mm × 10.2 mm, as shown on Figure 2.

The glue quantity of the PVAc and EPI adhesives has been chosen, according to technical data sheet from the manufacturer, as the best performing amount, and is reduced, compared to the other adhesives.

#### 2.2.2. X-ray Micro Computed Tomography

##### Equipment

The measurements were performed in the X-ray center of Empa Dübendorf, Switzerland on an EasyTom XL Ultra 230-160 (RX Solutions, Chavanod, France) using a L10711 160 kV microfocus tube (Hamamatsu, Japan), with a LaB6 cathode (limiting the voltage to 100 kV) and a 1 µm thick tungsten target, for the cooling reasons covered with 0.5 µm diamond and additional water-cooling of the tube head. The detector used was a Varian flat panel 2520 array, 1920 × 1536 pixels with 127 µm pitch with columnar grown CsI as the scintillating material. With a source-object distance of 16 mm and a source-detector distance of 340 mm, the resulting geometrical magnification was roughly 21. The used acceleration voltage was 70 kV with a nominal current of 70 µA and a resulting target current of 13 µA. The number of projections was 1440 over a 360° rotation of the sample. The scan time was roughly 0.5 h, using a frame rate of 3.5 Hz and an averaging of four frames. The heat transfer and the radiation dose on the sample can be neglected for this application.

For the 3D reconstruction (filtered back projection), the commercial software of RX solutions was used [9]. The filters applied were a ring filter (10 pixels), a phase filter (25%) and a reconstruction filter (75% sharpness). The voxel size of the reconstructed volume was 6 µm (unchanged from the projections).

##### Image Processing

The reconstructed data with the size of 1000 × 1300 × 1000 voxels were segmented in the industrial CT analysis software VGStudioMax3.3 (Volume Graphics GmbH, Heidelberg, Germany), using its advanced surface determination approach for a single material. The algorithm determines the material boundary, based on the local gray value differences considering the neighboring voxels. The surface determination was applied on a region of interest defining the area of the bonding. Based on the surface determination, a porosity analysis of the bonding area was performed, using the “only threshold” algorithm of VGStudioMax3.3. Pores with a volume smaller than 100 voxels, were filtered out to obtain the clear image and sort out disturbances. A probable threshold of 1.0 was used and the “check neighborhood” option was activated. Then, the pores detected in the interface between the wood and the adhesive, but belonging to the wood matrix, were deleted manually. The results of the porosity analysis were exported to Excel files.

#### 2.2.3. Nanoindentation

Determination of the indentation hardness (H_IT_—a measure of the resistance to permanent deformation or damage): the instrumented indentation test, according to DIN EN ISO 14577—nanoindentation.

The instrumented indentation test—often referred to as nanoindentation (NI) in many literature sources—is derived from the classic hardness test.

NI was initially used to determine the material properties of metals but has now also been used to a greater extent for plastics, especially in medicine and in the paint sector, among others [33,34,35,36].

The applied penetration force and the penetration path are measured continuously in the nano range (h ≤ 0.2 μm).

NI measurements are particularly suitable for characterizing elastic and plastic properties. It is used to determine the hardness, for example, of thin layers.

The contact area and consequently the hardness are calculated, depending on the geometry of the test tip, penetration force and penetration path.

The indentation hardness measurements were performed with indenter “Unat” (Zwick Roell, Ulm, Germany) with a pyramidal test tip (Berkovich) at the Institute for Materials Science at the TU Dresden.

The measurement method used the fast hardness-mode.

The experiments were performed in a load-controlled mode using a force of 0.4 mN. The load application time was 10 s, the hold time was 5 s with a force of 0.4 mN and unload time was 4 s to 0.054 mN.

The values were compared using the two-way analysis of variance (ANOVA).

The glue line thickness was determined using a digital microscope VHX-6000 (Keyence Corp., Osaka, Japan).

## 3. Results

### 3.1. X-ray Micro Computed Tomography

The volume fraction of the gaps (voids) in the adhesive joints was determined. The values represent the whole tested volume.

Table 4 contains the values of the total voids volume and their proportion, in relation to the total volume of the adhesive joint.

The highest gaps volume have PUR and PVAc adhesives. Then, decreasing, RPF, MF and EPI.

The voids are illustrated in Figure 3. The adhesive covers the whole bonded area, but in the joint, there were gaps, which are displayed on the figures. The size of the gaps is determined by the use of the colors displayed. The biggest voids are pink and the smallest are blue.

The results of the thickness determination, using a digital microscope are shown in Table 5.

The highest adhesive joint thickness was observed by the PUR and RPF adhesives. Then, decreasing, PVAc, MF, and EPI. The voids formation depends on various bonding parameters. One of the most important is an adhesive joint thickness. It was observed that the higher adhesive joint thickness correlated with the higher defect volume ratio (voids)—Figure 4. Adhesive joint thickness of PVAc is similar to MF, but differs in the defect volume ratio. It is probably caused by the lower solid content and the different curing method.

In the case of the PVAc and EPI adhesives, the reduced glue quantity was selected, based on the data sheet, as it is in industry use, in order to achieve the highest strength values.

### 3.2. Nanoindentation

The results of nanoindentation are shown in Table 6 and Table 7. Each variant was tested at 15 different points.

The highest values of hardness have condensation resins (RPF und MF). Then, decreasing, the EPI, PUR and PVAc adhesives.

In the investigations, two methods of measuring hardness were used and compared.

As shown in Figure 5, the two methods correlate with each other. Only for one variant (EPI) the method (adhesive layer on glass) showed a very large spread.

The selected 1C-PUR adhesives have a relatively low modulus of elasticity and hardness. The range of variation correlate with the values of Clauss. The hardness of some PURs is much higher, that parameter depends probably on the filler content [7].

For this reason, it is recommended to test the hardness of the adhesive applied to the wood.

Elastic modulus is used as a synonym for the reduced modulus E_r_.

The highest values of the elastic modulus have EPI, MF and then, decreasing, RPF, PVAc and PUR.

In the case of the elastic modulus, the method (adhesive layer on glass) showed also for the variant EPI, a much bigger spread. For this reason, the comparison of hardness and the elastic modulus was made, using the second method (adhesive on the wood)—Figure 6 and Figure 7.

This method is also less time consuming and requires less effort.

The results correlate with the literature data mentioned in Table 6 and Table 7.

## 4. Discussion

### 4.1. X-ray Micro Computed Tomography

In the case of the PUR adhesives, the CO_2_ gas bubbles are created during polymerization. During this process, water and moisture from the wood and the surrounding area are reacting with the adhesive.

PUR 2 has a higher proportion of gas bubbles, compared to PUR 1, due to a higher adhesive joint thickness.

In the case of the condensation resins, bubbles are formed during polycondensation, when water is released.

The PVAc adhesives are also cured physically by releasing water.

A high proportion of voids can significantly weaken the adhesive joint, because it creates deficiencies in a homogeneous adhesive joint. If the void is big enough to create a leak of adhesives between two wooden layers, it creates a weak spot.

The highest proportion of voids was observed by the PUR and PVAc adhesives.

It has been proven that the thickness of the adhesive joint and the type of adhesive have a large influence on the formation of voids in the adhesive joint. This has a significant impact on the final strength of the glued elements. Following the analysis of the results, it is recommended to achieve a joint thickness in the range of 100–150 µm for the appropriate application of the adhesive.

The variety of delamination is a result of both the specific nature of the substrate and the properties of the adhesive. These are the first research results in this area, and the authors are well aware of the complexity of the delamination—this paper uses an innovative method of analysis.

### 4.2. Nanoindentation

As shown, the PUR and PVAc adhesives show little variation. The examined methods of application to the adhesives differ slightly. In the case of the adhesive film applied to the glass in the case of the EPI adhesive, the standard deviation is quite large and for this reason this application method is not recommended any further. The highest hardness achieved is MF, then RPF and the EPI adhesive.

The modulus of elasticity of the films and the proportion of the voids are important for the shear strength (WFP) and delamination [33].

The MOE of the PUR adhesives could vary due to the chemical structure (crosslinking, fillers) [7].

It is known that the condensation resins, such as thermosets, have a higher hardness, but the EPI reached also high values, probably due to a very low proportion of voids in the adhesive joint. PUR adhesives have slightly higher values than the softest adhesive—PVAc.

Analyzing the glued samples, it can be concluded that lower glue application, as well as the higher pressing pressure and the proper surface preparation, can effectively lower the higher defect volume.

The preparation of the substrate surface, the content of the extractive substance on the surface and the surface tension, have a major impact on the curing of the adhesive and the formation of defects. This affects the hardness values and the e-modulus.

The glue joint should be thick enough to cover both bonded surfaces and penetrate them. However, it must not be too thick, as it will foam and there will be greater stress on the joint. For the best performance, the glue joint should be 0.1 mm [37].

## 5. Conclusions

The intention of the authors was to determine the suitability of a given innovative analytical method for evaluating the properties of adhesives. Therefore, the bonding agents representing all groups of adhesives were analyzed: polycondensation, polymerization and polyaddition. Based on the results obtained, it is possible to “modify” not only the properties of adhesives but the parameters of the adhesive processes.

The main findings are:The highest values of hardness have condensation resins (RPF und MF). Then, decreasing, the EPI, PUR and PVAc adhesives. The highest values of the elastic modulus have EPI, MF and then, decreasing, RPF, PVAc and PUR. The modulus of elasticity of the films and the proportion of the voids are important for the shear strength (WFP) and delamination.X-ray micro computed tomography can be used to create a 3D image of the glue joint and determine the proportion of the voids and their distribution in the adhesive joint. This proportion depends on the type of adhesive, glue quantity and reactivity. This proportion should be as reduced as possible.As a preparation for the evaluation of the hardness of adhesive films, it is recommended to apply the adhesives on wood, not glass. The MOE of the tested PUR is relatively low and can be increased by the chemical formation of the adhesive. At the same time, a higher WFP will be achieved.The chosen methods allow to efficiently evaluate the adhesive joints.

The following points should be further developed:The formation of voids depends strongly on the adhesive joint thickness. It would be interesting to investigate the precise influence of the adhesive joint thickness on the formation of voids.

## Figures and Tables

**Figure 1 materials-15-08604-f001:**
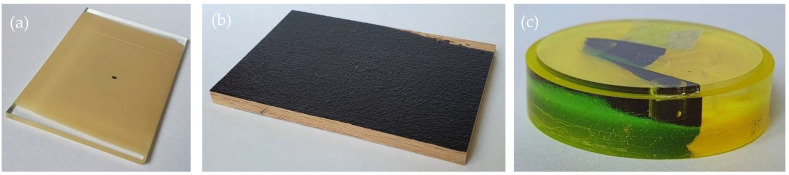
Prepared samples for the nanoindentation: (**a**) the adhesive film on glass, (**b**) on wood, (**c**) embedded in epoxy resin.

**Figure 2 materials-15-08604-f002:**
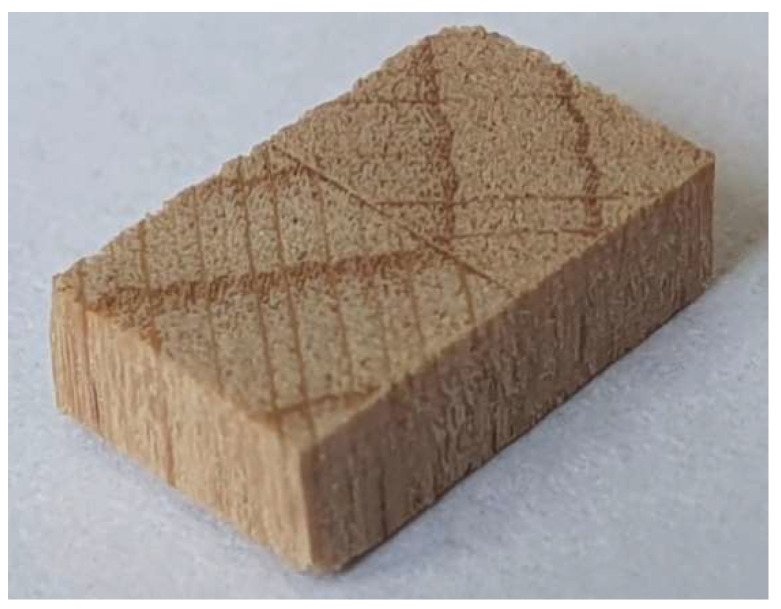
Prepared samples for the computer tomography.

**Figure 3 materials-15-08604-f003:**
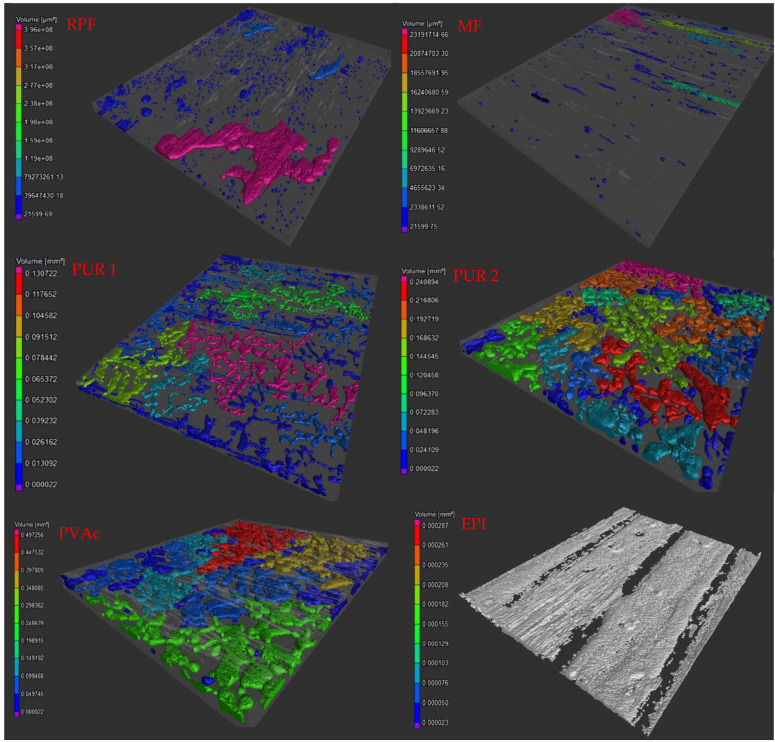
3D illustration of the voids in the adhesive joints.

**Figure 4 materials-15-08604-f004:**
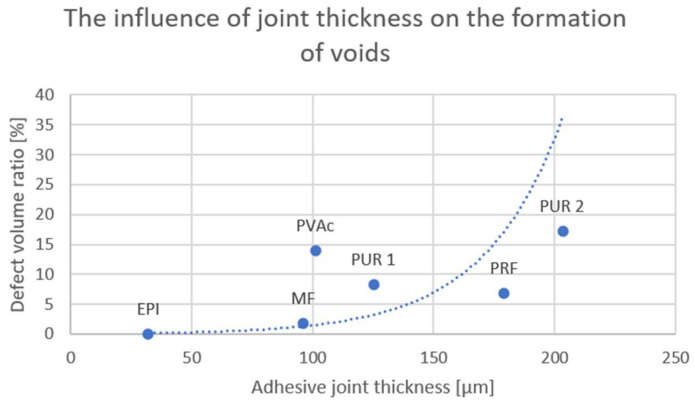
The influence of the joint thickness on the formation of voids.

**Figure 5 materials-15-08604-f005:**
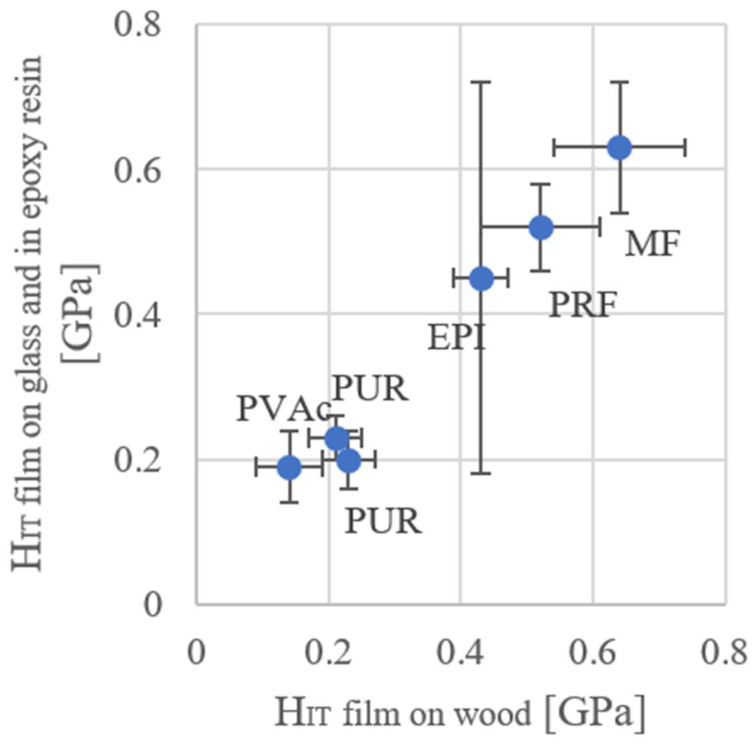
Comparison of the hardness from the nanoindentation of the pure polymer films and polymers in the adhesive on the wood (error bars correspond to the standard deviation).

**Figure 6 materials-15-08604-f006:**
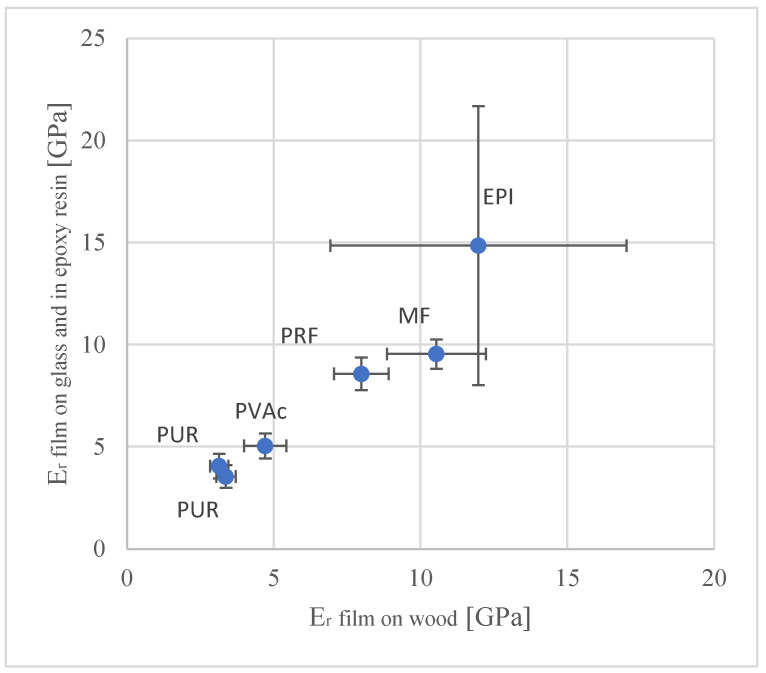
Comparison of the elastic modulus from the nanoindentation of the pure polymer films and polymers in the adhesive on the wood (error bars correspond to the standard deviation).

**Figure 7 materials-15-08604-f007:**
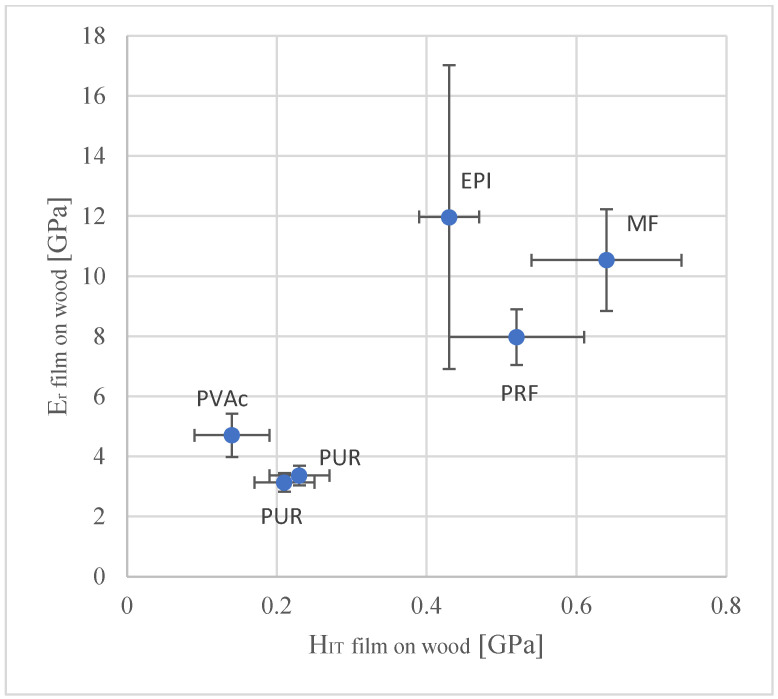
Mechanical properties of the wood adhesives, measured by means of nanoindentation (error bars correspond to the standard deviation).

**Table 1 materials-15-08604-t001:** Summary of the properties of beech wood, in comparison with the selected deciduous trees [32].

Wood Species		Beech (*Fagus sylvatica* L.)	Eiche (*Quercus robur* L.)	Esche (*Fraxinus excelsior* L.)
Extractive contents [%]	Benzene-alcohol extraction	1.5 … 1.9%	1.8 … 4.7%	4.4 … 5.4%
	Water extraction	1.9%	2.9 … 12.2%	1.4 …6.8%
Structural substances’ contents	Lignin content [%]	11.6 … 22.7%	24.9 … 34.3%	21.1 … 30.4%
	Total sugar content [%]	75.7 … 85.0%	73.2 … 73.2%	76.1 … 82.1%
	Cellulose content [%]	33.7 … 46.4%	37.6 … 42.8%	40.9 … 46.8%
Ash content [%]		0.3 … 1.2%	0.3 … 0.6%	0.43 … 0.61%
pH		5.1 … 5.4	3.9	5.8
Density (g/cm^3^) at 12% EMC		540 … 720 … 910 kg/m³	430 … 690 … 960 kg/m³	450 … 720 … 860 kg/m³
Differential volumetric shrinkage [%]		14.0 … 17.9 … 21.0%	12.6 … 15.6%	12.8 … 13.6%

**Table 2 materials-15-08604-t002:** List of adhesives and parameters used.

Adhesive	Glue Quantity [g/m^2^]	Pressing Time [min]	Pressing Temperature [°C]
RPF	180, both sides	300	20
MF	200, both sides	360	20
PUR 1	200, one-sided	80	20
PUR 2	200, one-sided	150	20
PVAc	185, one-sided	45	20
EPI	160, one-sided	45	20

**Table 3 materials-15-08604-t003:** Main physical properties of the used adhesives provided in the technical data sheets, according to the adhesive manufacturers.

Adhesive	Density [g/cm³]	Viscosity [mPas]	Solids Content [%]
RPF	1.15 ± 0.02	950 ± 550	58 ± 3
MF	1.21	12,500 ± 5300	64 ± 3
PUR 1	1.15	15,500 ± 2500	100
PUR 2	1.15	13,500 ± 2500	100
PVAc	1.05 ± 0.05	5000 ± 2000	49 ± 2
EPI	1.5 ± 0.05	11,000 ± 2000	60 ± 2

**Table 4 materials-15-08604-t004:** Proportion of the voids in the adhesive joints.

Adhesive	Defect Volume [mm^3^]	Defect Volume Ratio [%]
RPF	0.683	6.88
MF	0.083	1.78
PUR 1	0.758	8.31
PUR 2	2.291	17.26
PVAc	2.185	13.97
EPI	0.001	0.03

**Table 5 materials-15-08604-t005:** Penetration depths and adhesive joint thicknesses of the bonded test specimens.

Adhesive	Adhesive Joint Thickness [µm]	Penetration Depth [µm]
RPF	179 ± 6	115 ± 14
MF	96 ± 4	80 ± 4
PUR 1	125 ± 4	155 ± 25
PUR 2	203 ± 6	175 ± 12
PVAc	101 ± 5	20 ± 3
EPI	32 ± 2	30 ± 4

**Table 6 materials-15-08604-t006:** Hardness measured by means of nanoindentation (H_IT_) at 20 °C, of the adhesive films and layers, compared to the values from the literature [2].

Adhesive	H_IT_ of Adhesive Film on Beech Wood [GPa]	H_IT_ of Adhesive Layer in Epoxy Resin [GPa]	H_IT_ of Adhesive Layer on Glass [GPa]	H_IT_ Obtained by Clauss [GPa]
RPF	0.52 ± 0.09	0.52 ± 0.06	-	0.45 ± 0.02
MF	0.64 ± 0.10	0.63 ± 0.09	-	0.46 ± 0.01
PUR 1	0.23 ± 0.04	0.20 ± 0.04	-	0.16 ± 0.02
PUR 2	0.21 ± 0.04	0.23 ± 0.03	-	0.16 ± 0.02
PVAc	0.14 ± 0.05	-	0.19 ± 0.05	-
EPI	0.43 ± 0.04	-	0.45 ± 0.27	-

**Table 7 materials-15-08604-t007:** Elastic modulus measured by means of nanoindentation (E_r_) at 20 °C, of the adhesive films and layers.

Adhesive	E_r_ of Adhesive Film on Beech Wood [GPa]	E_r_ of Adhesive Layer in Epoxy Resin [GPa]	E_r_ of Adhesive Layer on Glass [GPa]	E_r_ Obtained by Clauss [GPa]
RPF	7.98 ± 0.93	8.57 ± 0.80	-	7.78 ± 0.39
MF	10.54 ± 1.69	9.54 ± 0.72	-	7.63 ± 0.76
PUR 1	3.37 ± 0.33	3.54 ± 0.55	-	3.03 ± 0.42
PUR 2	3.14 ± 0.31	4.05 ± 0.60	-	3.03 ± 0.42
PVAc	4.71 ± 0.72	-	5.04 ± 0.61	-
EPI	11.97 ± 5.05	-	14.85 ± 6.83	-

## Data Availability

Available from corresponding authors.

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
