# Peer review of "Investigations on the Characterization of Various Adhesive Joints by Means of Nanoindentation and Computer Tomography"

_materials, 2022, doi:10.3390/ma15238604_

Round 1
Reviewer 1 Report
In this work, the authors tested the mechanical properties of wood adhesive films cured in the dry state by nanoindentation. Further, the authors used X-ray microtomography to assess the quality of the bonded joints by determining the voids. The adhesives RPF, MF, PUR1, PUR2, PVAc and EPI were tested and analysed separately. This is an important and interesting study of the basic science at the interface between adhesives and wood, which will help to improve the bonding quality of wood products.
Q1: The thickness of the glue compound is an important factor in the quality of the glue compound. How do you think the thickness of the glue compound affects the glue defect volume ratio, please give a detailed example.
Q2: Adhesives with low solids content often improve the adhesive properties by increasing the thickness of the adhesive joint, on this basis how do you think a higher defect volume ratio can be avoided?
Q3: Please explain how the substrate affects the hardness and modulus of the adhesive in the test data for nanoindentation.
Q4: The authors provide a description of the experimental results, and I think it would improve the quality of the article for publication if the mechanisms were dissected in further detail.
Author Response
Dear Reviewer,
Thank you very much for your very interesting tips and comments!
I have taken them into account and they certainly improved the quality of our work.
In blue I have written back to the comments.
Review
In this work, the authors tested the mechanical properties of wood adhesive films cured in
the dry state by nanoindentation. Further, the authors used X-ray microtomography to
assess the quality of the bonded joints by determining the voids. The adhesives RPF, MF,
PUR1, PUR2, PVAc and EPI were tested and analysed separately. This is an important and
interesting study of the basic science at the interface between adhesives and wood, which
will help to improve the bonding quality of wood products.
Q1: The thickness of the glue compound is an important factor in the quality of the glue
compound. How do you think the thickness of the glue compound affects the glue defect
volume ratio, please give a detailed example.
The glue joint should be thick enough to cover both bonded surfaces and penetrate them.
But on the other hand, it must not be too thick, as it will foam and there will be greater
stress on the joint. For best performance, the glue joint should be 0.1 mm. [Kollman 1038-
1039 Band 2 1955]
Q2: Adhesives with low solids content often improve the adhesive properties by increasing
the thickness of the adhesive joint, on this basis how do you think a higher defect volume
ratio can be avoided?
Analyzing the glued samples, it can be concluded that lower glue application as well as
higher pressing pressure and proper surface preparation can effectively lower the higher
defect volume.
Q3: Please explain how the substrate affects the hardness and modulus of the adhesive in
the test data for nanoindentation.
The preparation of the substrate surface, the content of extractive substance on the surface
and the surface tension have a major impact on the curing of the adhesive and the formation
of defects. This affects the hardness values and e-modulus.
Q4: The authors provide a description of the experimental results, and I think it would
improve the quality of the article for publication if the mechanisms were dissected in further
detail.
RPF resin
The RPF resin is based on the reaction of resorcinol with formaldehyde. In the first stage, the
reaction yields linear chains. The addition of formaldehyde occurs preferably at positions 4
and 6 on the aromatic ring, while the position between the two hydroxyl groups is sterically
hindered. The reaction of resorcinol with formaldehyde is influenced by the molar ratio, the
concentration of the solution, the pH, the temperature and the catalyst types and alcohols
used.
MF resin
The basic reactions of MF production consist of methylolation and subsequent
condensation. The reaction of formaldehyde with the amino groups of melamine leads to
methylols, with a corresponding average degree of methylolation or distribution over the
individual methylolation stages depending on the formaldehyde excess.
PUR
PUR adhesives are based on polyadditions and polymerizations reactions. Due to an excess
of isocyanate, the reaction of isocyanate and polyols (polyester or polyatherpolyols)
produces chains with terminal and, if necessary, lateral isocyanate groups, which can react
with the moisture of the wood surfaces to be glued and thus lead to a cured system via this
addition reaction. Therefore, at least one of the two surfaces must supply the amount of
water required for curing, i.e. it must be porous and contain moisture.
During curing, the reaction of the isocyanate group with the moisture produces CO2, which
can cause foaming of the adhesive joint.
PVAc
PVAc adhesives are physically setting adhesives. The gluing effect of PVAc is based on the
removal of water by penetration into the surface or by evaporation.
EPI
The adhesive effect is achieved by evaporation of water from the glue line or absorption by
the parts to be joined (physical setting). In addition, chemical cross-linking takes place, which
leads to a significantly higher resistance to moisture and temperature. [3]

Reviewer 2 Report
See comments in the attached Review file. By transforming comments into text improvements, you will convey to the reader in a more perceptible form, both what has already been achieved before your research and the know-how added to your research.
I hope this works out very well for you and will give your readers ideas to join in developing new aspects of methods usage

Author Response
Dear Reviewer,
Thank you very much for your very interesting tips and comments!
I have taken them into account and they certainly improved the quality of our work.
In blue I have written back to the comments.
Review
Summary
At the beginning of the summary, it should be justified in a couple of sentences what are the
expected advantages of the nanoindentation and X-ray micro computed tomography methods over
the traditionally used ones. The advantage of these methods over traditionally used is their accuracy,
repeatability and lack of sample destruction. The samples are measured at the nanoscale. Thanks to
the fact that the samples are not destroyed during the test (as happens, for example, during strength
testing) they can be reproduced.
The purpose remains unclear: to rank the considered adhesives according to the average values of
the bonding properties provided by them or a comparative analysis of the indicators provided by the
two assessment tools and methodologies considered. The purpose is to do a comparative analysis of
the indicators provided by the two assessment tools and methodologies considered.
The full names of the tested adhesives should be mentioned once in full by adding their
abbreviations, using abbreviations below. In addition, the PVAC has modifications, it is necessary to
indicate which one is used in context. Done, PVAC wasn’t modified.
It is necessary to clarify the analysis of the void ratio of 0.03% obtained by the EPI, based on a more
thorough comparative analysis of the graphical information in the text, justifying to what extent the
resulting figure allows for unambiguous ranking of adhesives according to the obtained indicator.
Void ratio represents the ratio of the volume of voids to the total volume of the tested adhesive
bond.
1.Introduction
Consists of three somewhat unrelated in content subsections, as a result, each of them seems
unfinished and its connection to the subsequent chapters remains unclear. A brief summary at the
end of each subsection is necessary so that the reader better understands the need to evaluate
certain parameters of the adhesive joint and sees the need to use the two considered tools and
appropriate methodologies for interpreting the determined results. This would make it easier to
articulate the purpose of the study at the end of the chapter, which currently looks strongly
unrelated to the previous text. Corrected
Page 74-75: instead of modal need to use model. done
2. Materials and Methods
2.1.1. A reference to the said property of the beech wood (content of extractives) compared to other
varieties of deciduous trees should be added. As well experimental wood samples acclimatization
time should be added. done
2.1.2. The method of addition of the longitudinal polyamide fibers in PUR needs to be explained as it
can have a significant impact on the resulting adhesive properties (line)178-179. done
It is also desirable to justify the parameters of the variants of the adhesives summarized in the table
1, which are very different and may affect each of the properties discussed below in
different ways; at least with references from previous sources where they have been studied and
found suitable for the type of glue. done
Line 194-196 need a reference done
Need explanations of how to understand the viscosity values of adhesives in the Table 2? Or, for
example, a RPF binder used in the experiment with a viscosity in the range of 400-1510 mPas, which
seems technologically unacceptable; Or these are the boundaries within which the viscosity
indicators of existing producers fit in? Clarification needed. These values came from technical data
sheets. The range was surly smaller, but the exact values are not known.
2.2.2.2. and 2.2.2.3. subsections - it is desirable to add justifications for the choice of specific
parameters in relation to the object under study, for example, “Pores with a volume smaller than 100
voxels were filtered out” - why 100 voxels are chosen? That was chosen from an experience of
technician to get the clear image and sort out disturbances.
255-256: the geometry of the used test tip? done
Table 3 and Figure 3: Need clarification on what is shown in the ECI image - whether the adhesive
coating covers only part of the area and there are no voids (cavities) in it confirmed by the indicators
in the table, in turn, the part where there is no coating is not perceived by the method? The variety
of delamination is a result of both the specific nature of the substrate and the properties of the
adhesive. These are the first research results in this area, and the authors are well aware of the
complexity of delamination - this paper uses an innovative method of analysis.
But this, in turn, is captured by the nanoindentation method's information about the huge dispersion
in the ECI scores, such as Er 42% on wood (Table 6, Figure 5-7); consequently, the refusals of ECI
joints are not determined by the arithmetic mean, but by the minimum values of the indicator. On
the other hand, measurements may encourage changes in the adhesive composition and laying
technology in order to eliminate the problems identified by the advanced methods examined.
included
The obtained values can definitely help adhesive manufacturers formulate new adhesive systems.
A more detailed comparative analysis will allow for a more precise definition of the conclusions,
including a better understanding not only of the good aspects but also of the limitations of the use of
comparable methods. This will advance the experience of how to objectively determine the most
suitable adhesives with these tools according to the specifics of the wood to be joined.
This should be made clearer to the reader in the process of discussing the results and in the
summary.
The intention of the authors was to determine the suitability of a given innovative analytical method
for evaluating the properties of adhesives. Therefore, bonding agents representing all groups of
adhesives were analyzed: polycondensation, polymerization and polyaddition. Based on the results
obtained, it is possible to "modify" not only the properties of adhesives but the parameters of
adhesive processes.

Reviewer 3 Report
The topic of the research work and manuscript is really interesting, well-prepared and provides new information. However there are some issues to be addressed towards its quality improvement before publication. The key words should be enriched including more relevant simpler words, not complicated phrases. The acronyms GLT, CLT, NCO etc. should be explained the first time used in the text. In line 32, please provide as well the year of publication. In line 36, you could refer as well to the factor of wood elements surface quality before the glueing process (please provide some reference in this sentence). In line 40, a reference should be as well added. In line 43, what do you mean " price of components"? In line 44, replace the word "adequate" with "appropriate". In line 45, please provide the reference http://dx.doi.org/10.4067/S0718-221X2017005000008 to support your statement. In the paragraph of lines 50-58, there is not even one reference.The 1C-PUR is not explained the first time used in the text. The lines 59-61 could be part of the previous paragraph. The relevant work https://doi.org/10.1016/j.eurpolymj.2020.109690 could be used to support your statemennts. In line 67, please provide the year of publication. In line 92, correct the word "preformed". In line 96, replace "humidity" with "moisture content". In line 98, explain the "shear strength" (of what?). Combine the lines 100-104 together in the same paragraph. In line 107, the use of word "seperations" seems a little ackward (is it the right term?maybe mass discontinuities?). In line 120, check for syntactical errors that have been detected. Lines 123-124 can be combined with the previous lenes in the same paragraph. In line 127, reference should be added. Lines 130-144 combined in one paragraph. Line 141: sensibility should be changed to susceptibility. In lines 143-144, examples of those being less affected? The last paragraph of introduction is not very clear or specific, needs improvement in order to help the readers understand the exact objective of this work and the hypothesis of the whole experiment, as well as highlight the significance of this work. In 264 line , provide the manufacturer, country. Was there any statistical analysis applied in the results of this work? since it is not apparent and not at all described in materials-methods chapter. In conclusions chapter, you mainly summarize the findings, while the significance/meaning of this work is not highlighted. the results discussion section and conclusions are quite short and could be more extended and enriched and comprehenssive in terms of interpretation of the findingsAuthor Response
Dear Reviewer,
Thank you very much for your very interesting tips and comments!
I have taken them into account and they certainly improved the quality of our work.
In blue I have written back to the comments.
Review
The topic of the research work and manuscript is really interesting, well-prepared and
provides new information. However there are some issues to be addressed towards its
quality improvement before publication.
The key words should be enriched including more relevant simpler words, not complicated
phrases. wood gluing; nanoindentation; Micro CT; 1C-PUR; beech; wood adhesives; timber
construction;; adhesive joint; hardness; mechanical properties
The acronyms GLT, CLT, NCO etc. should be explained the first time used in the text
NCO (Isocyanate), GLT (Glue-laminated timber), CLT (Cross Laminated Timber)
In line 32, please provide as well the year of publication. 2021
In line 36, you could refer as well to the factor of wood elements surface quality before the
glueing process (please provide some reference in this sentence). I added surface quality as
parameter and the reference [1] refers to the entire paragraph.
In line 40, a reference should be as well added. The reference [1] refers to the entire
paragraph.
In line 43, what do you mean " price of components"? Price of hardwood is higher than price
of softwood. That is the main reason why hardwood is not so popular.
In line 44, replace the word "adequate" with "appropriate". done
In line 45, please provide the reference http://dx.doi.org/10.4067/S0718-
221X2017005000008 to support your statement. done
In the paragraph of lines 50-58, there is not even one reference. The 1C-PUR is not explained
the first time used in the text. done
The lines 59-61 could be part of the previous paragraph. The relevant
work https://doi.org/10.1016/j.eurpolymj.2020.109690 could be used to support your
statements. done
In line 67, please provide the year of publication. done
In line 92, correct the word "preformed". done
In line 96, replace "humidity" with "moisture content". done
In line 98, explain the "shear strength" (of what?). of bonded elements
Combine the lines 100-104 together in the same paragraph. In line 107, the use of word
"seperations" seems a little ackward (is it the right term? maybe mass discontinuities?).
done
In line 120, check for syntactical errors that have been detected. done
Lines 123-124 can be combined with the previous lenes in the same paragraph. done
In line 127, reference should be added. done
Lines 130-144 combined in one paragraph. done
Line 141: sensibility should be changed to susceptibility. done
In lines 143-144, examples of those being less affected? e.g. condensation resins
The last paragraph of introduction is not very clear or specific, needs improvement in order
to help the readers understand the exact objective of this work and the hypothesis of the
whole experiment, as well as highlight the significance of this work. The purpose of this
study was to use commercially available methods (NI and used X-ray micro computed
tomography) and find a way to evaluate the adhesive bond and observe the relationship for
various commonly used wood adhesives.
It is expected that especially PUR adhesives, due to their known foaming, will have a lot of
voids in the glue joint.
In 264 line , provide the manufacturer, country. done
Was there any statistical analysis applied in the results of this work? since it is not apparent
and not at all described in materials-methods chapter. The values were compared using the
two-way analysis of variance (ANOVA).
In conclusions chapter, you mainly summarize the findings, while the significance/meaning
of this work is not highlighted. the results discussion section and conclusions are quite short
and could be more extended and enriched and comprehensive in terms of interpretation of
the findings. done

Round 2
Reviewer 3 Report
As I have checked the authors have implemented the proposed changes in the revised verion of manuscript towards the improvement of their work. Almost all the changes have been implemented and in my opinion, the manuscript is well-prepared and organized enough to be accepted for publication in this journal. I remain at your disposal for any clarification.